# An Analysis of the Decidability and Complexity of Numeric Additive Planning

**Primary Keywords:** *Theory*

## Abstract

In this paper, we first define numeric additive planning (NAP), a planning formulation equivalent to Hoffmann's Restricted Tasks over Integers. Then, we analyze the minimal number of action repetitions required for a solution, since planning turns out to be decidable as long as such numbers can be calculated for all actions. We differentiate between two kinds of repetitions and solve for one by integer linear programming and the other by search. Additionally, we characterize the differences between propositional planning and NAP regarding these two kinds. To achieve this, we define so-called multi-valued partial order plans, a novel compact plan representation. Finally, we consider decidable fragments of NAP and their complexity.

**Keywords**: Action based planning · Numeric planning · Planning as search · Integer linear programming.

## Introduction

In classical propositional planning scenarios, effects are *activations/deactivations* of certain propositions, i.e., assignments of truth values $\top, \bot$. Notice that activating $p$ twice instead of once results in the same state, and, ignoring preconditions, any two (propositional) plans of the form $[a, b, c], [a, b, b, c]$ are equivalent, but $[b, a, c]$ is not equivalent to $[a, b, c]$ in general, i.e., repeating an action without changing the order does not change a propositional plan.

Conversely, order could be irrelevant in some numeric planning scenarios. For example, if actions have no preconditions and can only increase or decrease variables by a constant, then it is only relevant to know how often each action should be performed in the plan irrespective of order. Effects in this setting are *additive*. Since addition is associative, we can assume that all additions occur in parallel. This problem is an essential part of numeric planning, where additive effects are allowed, but actions could also have preconditions, i.e., where both order and the number of parallel additive effects are relevant.

To illustrate the difference, consider the effects of actions like *pay_ten_euros* and *open_the_door*. One can immediately see the difference when one imagines a plan containing two successive repetitions of each action. If a minimal plan begins with the action *open_the_door*, there is no need to repeat this action later on in that plan, as long as none of its effects is deactivated by a following action, like *close_the_door*.

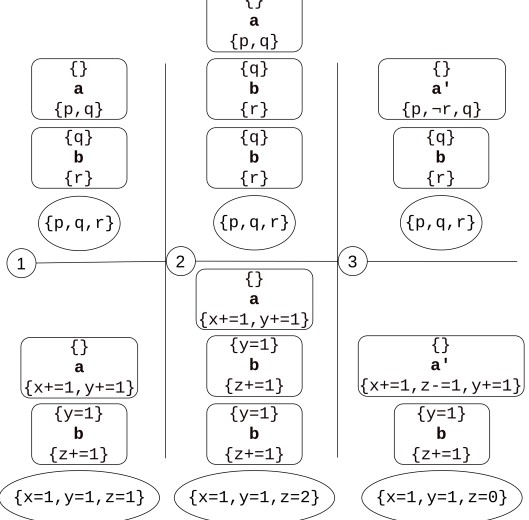

Figure 1: Demonstration of the expressive power of additive (below) compared to de-/activation effects (above). Compare $p, q, r$ with $x, y, z$, respectively, in all actions (rectangles) and the resulting states (ellipses) for all three plans.

However, one can easily imagine two successive repetitions of *pay_ten_euros*, without any of its effects getting "deactivated" or "reversed". One could perform these two actions in parallel via *pay_twenty_euros*. This looks intuitive here, but it becomes hard to know which actions can be done in parallel when the problem is more complicated, e.g., if *pay_ten_euros* has preconditions that make it not always repeatable twice in a row.

To finalize our observations, we show in Fig. 1 six plans, three propositional (above), and three numeric (below). All actions are described by their preconditions above and effects below the action name. By $x{+}{=}1$ we mean that the action increments $x$ by 1 after applying it, and $x{-}{=}1$ means that the value is decremented by 1. Initially, $p, q, r$ are deactivated (false), and $x = y = z = 0$. Compare the plans from left to right, where (1) on the left, we begin with a simple plan; (2) in the middle, we repeat the action $b$; and (3) on the right, we add a negative effect to $a$. Notice that all preconditions and effects have the same overall structure w.r.t. the

variables $p$ with $x$, $q$ with $y$, and $r$ with $z$. We can see that all three propositional plans result in the same state, while the numeric plans result in three different states. These examples illustrate the additional expressive power of numeric additive over activation/deactivation effects.

In a nutshell, we say that in the plan $[a, \underline{b}, \underline{b}, c]$, there are two *parallel* repetitions of $b$, and in the plan $[a, \underline{b}, d, \underline{b}, c]$, there are two *sequential* repetitions of $b$, because they are separated by an occurrence of $d$. As discussed before, no parallel repetitions of propositional actions like *open_the_door* are relevant. However, numeric planning is a superset of propositional planning, i.e., parallel and sequential repetitions of actions could be needed for the solution. This paper aims to study the differences between these two kinds and find a compact and least committing representation for both. This will help structure the search for solutions to the planning problem and lead to novel decidable fragments.

**Paper outline:** We define the planning problem and study its expressivity in Section 3. Then, in Section 4, we define a new way of representing plans. In Section 5, we introduce the solving algorithm. Finally, we define three decidable fragments in Section 6.

## Related Work

Introducing unbounded numeric variables makes planning problems undecidable in general (Helmert 2002). In recent years, many results have been published to tackle this issue by introducing restrictions on the planning problem: (Shin and Davis 2005; Gerevini, Saetti, and Serina 2008; Eyerich, Mattmuller, and Roger 2009; Coles, Fox, and Long 2013; Scala et al. 2017; Aldinger and Nebel 2017; Li et al. 2018; Piacentini et al. 2018; Kuroiwa et al. 2021; Shleyfman, Kuroiwa, and Beck 2023; Gnad et al. 2023). Specifically, Helmert (2002) presents decidable cases based on restricting the allowed mathematical functions and relations. We continue his work and identify decidable fragments based on their structural properties, similar to the causal graphs approach (CG) (Jonsson and Bäckström 1998; Brafman and Domshlak 2003; Helmert 2004; Giménez and Jonsson 2008). We consider a variant of simple numeric planning (SNP) (Scala et al. 2016), which is equivalent to restricted tasks (RT) (Hoffmann 2003). Shleyfman et al. (2022; 2023) show how RT/SNP with restrictions on the causal structure of variables is in PSPACE. We extend this idea by focusing on the causal structure of actions. Additionally, many complexity results are considered w.r.t. total or partial order plans, while we use the ability for some exponential length plans to be described with a polynomial-size representation.

Furthermore, many results on the advancements of planning solvers show how numeric planning could benefit from heuristic search, e.g., by using abstraction methods: (Seipp and Helmert 2014; Illanes and McIlraith 2017, and others). We follow the line of Hoffmann, who (2003) proposed an extension of delete lists, a relaxation of propositional planning, to numeric planning. We will extend his ideas about relaxed planning graphs (RPG) by ignoring "violations". Last but not least, we base our approach on some of the insights from LP-RPG (Coles et al. 2008) about the number of occurrences of an action in a plan, but we will use more general terms like sequential and parallel repetitions.

## Numeric Additive Planning (NAP)

The following definition will allow for a straightforward translation of the planning problem into an ILP system, as we will see in Section 4. Although, at first glance, the formulation lacks propositional preconditions and effects, we will include these later. We use the Greek letter $\pi$ for preconditions and $\sigma$ for the additive (sum) effects.

**Definition 1.** *A numeric additive planning domain (denoted NAD) is a tuple $\mathcal{A} = (A, V, \sigma, \pi)$, where $A, V$ are finite nonempty sets of actions and variables, respectively. Additionally, $\sigma : A \times V \to \mathbb{Z}$ defines the additive effect of each action on each variable, and $\pi : A \times V \to \mathbb{Z} \cup \{-\infty\}$ defines the numeric precondition of each action for each variable.*

For each action $b \in A$ and each variable $x \in V$, it is meant for the current value of $x$ to be greater than or equal to $\pi(b, x)$ for $b$ to be applicable. If $\pi(b, x) = -\infty$, then the preconditions of $b$ are not dependent on the value of $x$. After applying $b$, the current value of $x$ is incremented/decremented by the (constant) integer value $\sigma(b, x) \in \mathbb{Z}$. If $\sigma(b, x) = 0$, the action $b$ has no effect on variable $x$. We say $[x \geq v]$ is a precondition of $b$ if $\pi(b, x) = v$, and $[x{+}{=}v]$ is an effect of $b$ if $\sigma(b, x) = v$ as well as $[x{-}{=}v]$ if $\sigma(b, x) = -v$, for $v \in \mathbb{Z}$.

**Definition 2.** *For a NAD $\mathcal{A} = (A, V, \sigma, \pi)$, we define a state $S : V \to \mathbb{Z}$ as a function assigning an integer value to each variable. Additionally:*

- *For an action $a \in A$, a state satisfies the preconditions of $a$ iff $S(x) \geq \pi(a, x)$ for all $x \in V$ (denoted $S \vDash a$).*
- *An action $a \in A$ can be added to a state by adding its effects, i.e., $(S + a)(x) := S(x) + \sigma(a, x)$ for $x \in V$.*

With states defined, we are ready to define the validity of any sequence of actions.

**Definition 3.** *For $n \in \mathbb{N}_{\geq 1}$, and actions $a_1, ..., a_n \in A$, we call the sequence of actions $l := [a_1, a_2, ..., a_n]$ a plan. We denote the length of the plan with $|l| = n$, and say that $l$ is valid from an initial state $S_0 : V \to \mathbb{Z}$ if $S_i \vDash a_i$, with $S_i := S_{i-1} + a_{i-1}$ if $i \geq 1$, for all $i \in \{0, .., n\}$.*

In this paper, unlike many other planning formulations, we will encode the set of goal states as a precondition of one of the actions (usually named $g$). In other words, we ask the following question: Given an initial state $S_0$ and an action $g$, does a valid plan that ends with $g$ exist? I.e., is it possible to reach a state $S$ s.t. $g$ is applicable ($S \vDash g$)? Therefore, we will denote the planning problem as $\{S_0 \to g\}_{\mathcal{A}}$.

**Definition 4.** *For a NAD $\mathcal{A} = (A, V, \sigma, \pi)$, an initial state $S_0 : V \to \mathbb{Z}$, and an action $g \in A$, a numeric additive planning problem (denoted NAP) $\{S_0 \to g\}_{\mathcal{A}}$ is defined as the set of plans $l = [a_0, ..., a_n]$, s.t. $a_n = g$ and $l$ is valid from $S_0$.*

If the context is clear, we will use $A, V, \sigma, \pi$ without explicitly referring to that $\mathcal{A} = (A, V, \sigma, \pi)$, when speaking of NAP $\{S_0 \to g\}_{\mathcal{A}}$.

### Expressiveness

**Preconditions:** In NAP, we only allow preconditions of the form $[x \geq v]$ for a variable $x \in V$ and an integer $v \in \mathbb{Z}$. However, for any precondition of the form $[x \leq v]$, we can define a new variable $\overline{x}$ s.t. for all $a \in A$, $\sigma(a, \overline{x}) = -\sigma(a, x)$. This implies that if the initial state $S_0 : V \to \mathbb{Z}$, respects $S_0(\overline{x}) = -S_0(x)$, then, any reachable state $S : V \to \mathbb{Z}$ from $S_0$, we get $S(\overline{x}) = -S(x)$. I.e., $[x \leq v]$ is equivalent to $[\overline{x} \geq -v]$, which is representable in NAP. We can, therefore, represent preconditions of the form $[x = v]$ as $[x \geq v] \wedge [\overline{x} \geq -v]$. Additionally, preconditions of the form $[x > v]$ can be represented by $[x \geq v+1]$, since we only deal with integers. Finally, if an action $a$ has a disjunction in its preconditions, e.g., $[x \neq v] \equiv [x > v] \vee [x < v]$, we can create two actions $a_1, a_2$ with the same effects but $a_1$ has $[x > v]$ as its precondition and $a_2$ has $[x < v]$. Finally, for an action $a$ with the conditional effect $[x \geq v] \to [y \mathrel{+}= w_1] \wedge [x < v] \to [y \mathrel{+}= w_2]$, for $x, y \in V, v, w_1, w_2 \in \mathbb{Z}$, we can define two actions $a_1, a_2$ s.t. $a_1$ is applied in the first situation (precondition $[x \geq v]$ and effect $[x \mathrel{+}= w_1]$), and $a_2$ is applied in the second situation. It is also important to mention that preconditions with a linear combination of variables, e.g., $[x + 2y - z \geq v]$, can be represented in NAP by defining a new variable $t := x + 2y - z$ and ensuring that all the effects on $x, y, z$ are combined with effects on $t$ with a factor $1, 2, -1$ respectively. All these transformations can be easily implemented and reversed. For this reason, in NAP, we can focus only on conjunctions of preconditions of the form $[x \geq v]$ and unconditional additive effects.

**Effects:** NAP lacks the de-/activating effects used in propositional planning (PP). However, this does not necessarily mean that PP cannot be reduced to NAP in polynomial time. Given a PP domain $\mathcal{A} := (A, P, \textit{eff}, \textit{pre})$ with sets of actions and propositions $A, P$, respectively. We define $Lit := \{p, \neg p : p \in P\}$ as the set of literals. Finally, $\textit{pre}, \textit{eff} : A \to 2^{Lit}$ are functions defining the preconditions and effects of each action. A state in PP is a function $S : P \to \{\top, \bot\}$. A PP problem is given as a tuple $(\mathcal{A}, I, G)$, where $I$ is the initial state and $G \subseteq Lit$ describes a set of goal states $S$ s.t. $S(p) = \top$ iff $p \in G$, and $S(p) = \bot$ iff $\neg p \in G$.

For each proposition $p$, we create a numeric variable $x_p$. For an initial state $I$, we create an initial state $S_I$, s.t. $S_I(x_p) = 1$ iff $I(p) = \top$, and $S_I(x_p) = 0$ iff $I(p) = \bot$. For each propositional precondition $p$, we add a numeric precondition $[x_p = 1]$, and for $\neg p$, we add $[x_p = 0]$. And finally, for each propositional effect $p$, we add a numeric effect $[x_p \mathrel{+}= 1]$ and for $\neg p$, we add $[x_p \mathrel{-}= 1]$. Notice that in PP, as discussed in the introduction, an action $a$ with a precondition $p$ can be applied even if $p$ was activated twice before, but this does not work for our current translation. E.g., for a proposition $r$, propositional actions $a, b, g$, and initial state $I$, if $I(r) = \bot$, $r \in \textit{eff}(b) \cap \textit{pre}(g)$, and $\neg r \notin \textit{eff}(a)$, then, if $[a, b, b, g]$ is valid from $I$ in PP, $[a, b, b, g]$ is not valid from $S_I$ in NAP with the translation above because the state $S$ before applying $g$ has $S(x_r) = 2$, but $g$ has the precondition $[x_r = 1]$, compare the middle plans in Fig. 1. Therefore, for each proposition $r$, we need two new actions $c_r, c_{\neg r}$ that cor-

rect the numeric value representing a proposition $r$ from any value greater than 1 to 1 and from any value less than 0 to 0, i.e., $c_r$ has the precondition $[x_r > 1]$ and effect $[x_r \mathrel{-}= 1]$, and $c_{\neg r}$ has the precondition $[x_r < 0]$ and effect $[x_r \mathrel{+}= 1]$. Now, if $[a, b, b, g]$ is valid from $I$ in PP, then, $[a, b, b, c_r, g]$ is valid from $S_I$ in NAP, where $c_r$ corrects one of the two activations of $r$ caused by the two repetitions of $b$ before $a$. Therefore, any PP domain $\mathcal{A} = (A, P, \textit{eff}, \textit{pre})$ can be translated to a NAD with $|A| + 2|P|$ actions and $2|P|$ variables.

### Complexity

We study the complexity of NAP w.r.t. $\|A\|$, the minimal number of bits required to encode a NAD $\mathcal{A}$.

**Theorem 1.** *Hoffmann's Restricted Tasks (RT) can be reduced to* NAP *in polynomial time.*

*Proof.* NAP is similar to RT, except that it does not allow for propositional preconditions, which were shown to be included in NAP without exponential blowup. Additionally, we can transform any NAP over rationals into a NAP over integers by scaling all variables with suitable factors. Parts of this proof are discussed by Hoffman (2003). $\square$

Therefore, NAP is undecidable as it can model Abacus programs, as Helmert (2002) shows. In that paper, a NAP problem is equivalent to PLANEX-$(\mathcal{C}_c, \mathcal{C}_c, \mathcal{E}_{\pm c})$. Gnad et al. (2023) show that multiplication and division by constant and general assignment effects are included in RT/-NAP, proving the undecidability by reducing Collatz problems to it.

### Example

An investment company starts with an initial capital $c_0 \in \mathbb{N}$ and can perform buying and selling actions to reach a goal profit of $p \in \mathbb{N}$. Let us model a small example of this problem as NAP: We define the variables $c$ for the current capital and $q$ for the number of products in stock. Additionally, two actions: $b$ for buying one instance of a product for 4 units of money, $\sigma(b, c) = -4$, preconditioned by having this amount of money, $\pi(b, c) = 4$, and $s$ for selling it with 25% profit, $\sigma(s, c) = 5$. In a simpler notation:

- The action $b$, to buy, with preconditions $\{[c \geq 4]\}$, and effects $\{[c \mathrel{-}= 4], [q \mathrel{+}= 1]\}$.
- The action $s$, to sell, with preconditions $\{[q \geq 1]\}$, and effects $\{[c \mathrel{+}= 5], [q \mathrel{-}= 1]\}$.
- The goal action $g$, with preconditions $\{[c \geq c_0 + p]\}$.

Finally, the initial state is defined by $S_0(c) := c_0$ and $S_0(q) := 0$. E.g., if $p = 2$ and $c_0 = 8$, then $l_1 = [b, b, s, s, g]$ and $l_2 = [b, s, b, s, g]$ are both valid plans. However, if the initial capital $c_0 = 7$, then only $l_2$ is valid because there is not enough capital to buy twice in a row before selling once. This example will be used (always in *italic*) to demonstrate the algorithms discussed in this paper because of its simplicity. Of course, any sophisticated investment process involves more actions than buying and selling the same product. However, the results of this paper generalize to many other planning domains with complicated dependencies.

## Multi-Valued Partial Order Plans (MvPOPs)

Let us first discuss the idea of a multi-valued relation. For example, a classical binary relation $r$ over a set $K$ is a subset of $K^2$. We can represent this relation by $f_r : K^2 \to \{0, 1\}$, s.t. $f_r(b, s) = 1$ iff $(b, s) \in r$. We extend the concept by allowing values other than 0 and 1, e.g., $f_r : K^2 \to \mathbb{N}$. We usually represent solutions in planning as a partial order relation $<$. An action $b$ is ordered before $s$, denoted $b < s$, here $f_<(b, s) = 1$, if $b$ occurs before $s$. With $f_<(b, s) = 2$ we can represent that $b$ occurs twice before $s$. We will use this representation for the number of needed *parallel* repetitions of actions. In cases where $b$ occurs once before $s$ and once after it as in $l_2 = [b, s, b, s, g]$, we define $b'$ to be a new action with the same preconditions and effects as $b$. $l_2$ is thus equivalent to $l'_2 = [b, s, b', s, g]$, where $b, b'$ are *sequential* repetitions of $b$. We can alleviate the need for labeling different sequential repetitions (e.g., via $b, b', b'', ...$, or $b_1, b_2, b_3, ...$) by using multi-sets of actions $\mathbf{A} = (A, \mu)$ for $\mu : A \to \mathbb{N}$, where $\mu(a)$ is equal to the number of *sequential* repetitions of action $a$. Let $|\mathbf{A}| := \sum_{a \in A} \mu(a)$. We use matrices $\mathfrak{P} \in \mathbb{N}^{|\mathbf{A}| \times |\mathbf{A}|}$ to describe the multi-valued relation $f_<$ mentioned before. Instead of enumerating the actions to match them with rows and columns, we label the matrix entries by their respective actions directly, where $f_<(a, b) = \mathfrak{P}[b, a]$, and $\mathfrak{P}[b, a]$ is the entry in the row of $b$ and column of $a$ in the matrix $\mathfrak{P}$, and it represents the number of *parallel* repetitions of $a$ before $b$.

*For the plan $l_2 = [b, s, b, s, g]$. Given the (unordered) multi-set $\mathbf{A}_2 = \{g, s, b, s, b\}$, the MvPOP $\mathfrak{P}_2$ over $\mathbf{A}_2$ with max. $g$ described by the following matrix, is an equivalent representation to $l_2$ (Fig. 2):*

| $\mathfrak{P}_2$ | $g$ | $s$ | $b$ | $s$ | $b$ |
|---|---|---|---|---|---|
| $g$ | 0 | 1 | 1 | 1 | 1 |
| $s$ | 0 | 0 | 1 | 1 | 1 |
| $b$ | 0 | 0 | 0 | 1 | 1 |
| $s$ | 0 | 0 | 0 | 0 | 1 |
| $b$ | 0 | 0 | 0 | 0 | 0 |

*Notice the two different rows and columns for $b$. For example, $\mathfrak{P}_2[b, s] = 1$ (3rd row, 4th column) corresponds to the occurrence of the first sequential repetition of $b$ before $s$, and $\mathfrak{P}_2[s, b] = 1$ (2nd row, 3rd column) corresponds to the occurrence of $s$ before the second sequential repetition of $b$.*

On a side note, for any MvPOP $\mathfrak{P}$ over $\mathbf{A}$, an ordering of $\mathbf{A}$ must exist, s.t., the matrix of $\mathfrak{P}$ is triangular because otherwise, the represented relation contains a cycle.

**Definition 5.** *An MvPOP $\mathfrak{P}$ over a multi-set $\mathbf{A} = (A, \mu)$ is a matrix in $\mathbb{N}^{|\mathbf{A}| \times |\mathbf{A}|}$ s.t. for all $a, b, c \in \mathbf{A}$:*

- *Irreflexive: $\mathfrak{P}[a, a] = 0$.*
- *Asymmetric: $\mathfrak{P}[b, a] > 0$ implies $\mathfrak{P}[a, b] = 0$.*
- *Transitive: $\mathfrak{P}[c, b] > 0$ implies $\mathfrak{P}[c, a] \geq \mathfrak{P}[b, a]$.*

*We say that $g \in \mathbf{A}$ is a maximum (denoted max.) of $\mathfrak{P}$ iff $\mathfrak{P}[a, g] = 0$ for all $a \in \mathbf{A}$.*

Classically, partial order plans are a compact representation of a set of plans, known as *linearizations*, *linear extensions*, or *total ordering* of the partial order. For example, if

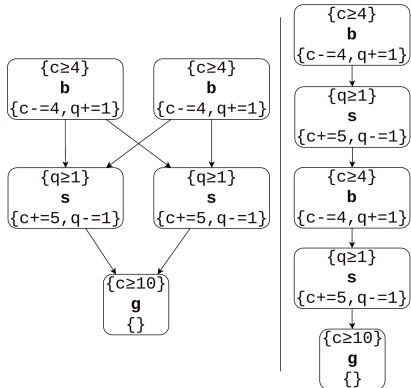

Figure 2: Partial orders representing $\mathfrak{P}_1$(left) and $\mathfrak{P}_2$(right).

$a < b$, then $a$ should occur before $b$ in any linearization of $<$. However, in an MvPOP $\mathfrak{P}$, there might be multiple occurrences of an action $a \in \mathbf{A}$. For this reason, we will assume that a linearization of $\mathfrak{P}$ must have at least $\mathfrak{P}[b, a]$ occurrences of $a$ before the first occurrence of $b$ for all $a, b \in \mathbf{A}$.

Let $\#_a(l)$ be the number of occurrences of action $a$ in a plan $l$, and let $l[..b]$ be the sub-plan of $l$ from the first action to the first occurrence of $b$. E.g., if $l = [a, c, c, b, e, b, g]$, then $l[..b] = [a, c, c, b]$. Notice that in any MvPOP, if $g$ is the only maximum, it occurs at the end of any linearization. Additionally, the number of occurrences of an action $a \neq g$ should be fixed $\#_a(l) = \mathfrak{P}[g, a]$ for all linearizations $l$ of $\mathfrak{P}$. Therefore, and because of transitivity, $l[..b]$ must be contained in $l[..g]$, i.e., $\mathfrak{P}[g, a] \geq \mathfrak{P}[b, a]$. We can thus define the number of occurrences of $a$ before $b$ to be between $\mathfrak{P}[b, a]$ and $\mathfrak{P}[g, a]$ in any linearization, for all $a, b \in \mathbf{A}$

**Definition 6.** *The set of linearizations of an MvPOP $\mathfrak{P}$ (denoted $\text{lin}(\mathfrak{P})$) over $\mathbf{A} = (A, \mu)$ with exactly one maximum $g$ is the set of plans $l \in A^*$ s.t. $\mathfrak{P}[b, a] \leq \#_a(l[..b]) \leq \mathfrak{P}[g, a]$ for all $a, b \in \mathbf{A}$, and $g$ occurs once at the end of $l$.*

### Multi-Valued Total Order Plans

We call a multi-valued partial order with exactly one linearization a multi-valued total order plan (MvTOP).

**Lemma 1.** *For any plan $l = [a_0, ..., a_n]$ there exists a multi-set $\mathbf{A} = (A, \mu)$ and an MvTOP $\mathfrak{P}_l$ over $\mathbf{A}$ with max. $a_n$ s.t. $\text{lin}(\mathfrak{P}_l) = \{l\}$.*

*Proof.* Let $\mu(a) := \#_a(l)$, $\mathfrak{P}_l[a_j, a_i] = 1$ if $i < j$, and $\mathfrak{P}_l[a_j, a_i] = 0$ else, for $a_i, a_j \in \mathbf{A}$. Then, $\text{lin}(\mathfrak{P}_l) = \{l\}$. $\square$

Next, we will show, through an example, an illustration of the translation of the NAP validity checking of an MvPOP into an ILP satisfiability problem by constraining the states that are produced through the plan.

*Given $l_1 = [b, b, s, s, g]$. Let $\mathfrak{P}_1$ be an MvPOP over $\{g, s, b\}$ with max. $g$, s.t. $\text{lin}(\mathfrak{P}_1) = \{l_1\}$ (Fig. 2):*

| $\mathfrak{P}_1$ | $g$ | $s$ | $b$ |
|---|---|---|---|
| $g$ | 0 | 2 | 2 |
| $s$ | 0 | 0 | 2 |
| $b$ | 0 | 0 | 0 |

For a multi-set $\mathbf{A} = (A, \mu)$ and a NAD $\mathcal{A} = (A, V, \sigma, \pi)$, we define the matrices that represent the effects and preconditions, respectively, $\Sigma, \Pi \in (\mathbb{Z} \cup \{-\infty\})^{|\mathbf{A}| \times |V|}$, with $\Sigma[a, x] := \sigma(a, x), \Pi[a, x] := \pi(a, x)$, for all actions $a \in \mathbf{A}$ over all variables $x \in V$. Finally, the matrix $S_0 \in \mathbb{Z}^{|\mathbf{A}| \times |V|}$ represents an initial state $S_0$, by $S_0[a, x] := S_0(x)$ for all $a \in \mathbf{A}, x \in V$.

*In our example, $\Sigma$ and $\Pi$ over $\mathbf{A} = \{g, s, b\}$ is:*

| $\Sigma$ | $c$ | $q$ | | $\Pi$ | $c$ | $q$ |
|---|---|---|---|---|---|---|
| $g$ | $0$ | $0$ | | $g$ | $10$ | $-\infty$ |
| $s$ | $+5$ | $-1$ | | $s$ | $-\infty$ | $1$ |
| $b$ | $-4$ | $+1$ | | $b$ | $4$ | $-\infty$ |

*Now, for an initial capital $c_0 = 8$, if $S := S_0 + \mathfrak{P}_1 \Sigma$, then:*

$$
\begin{aligned}
S[s, q] &= 0 + 0 \cdot \sigma(s, q) + 2 \cdot \sigma(b, q) \\
&= 0 + 0(-1) + 2(+1) = 2 \geq 1 = \Pi[s, q] \\
S[g, c] &= 8 + 2 \cdot \sigma(s, c) + 2 \cdot \sigma(b, c) \\
&= 8 + 2(5) + 2(-4) = 10 \geq 10 = \Pi[g, c]
\end{aligned}
$$

*These are the values of the variables $q, c$ before applying the actions $s, g$, in any linearization of $\mathfrak{P}_1$, respectively, which satisfy the precondition of both action on the respective variable, ensuring the plan's validity (compare Fig. 2). We generalize this result.*

**Lemma 2.** *If $l \in \{S_0 \to g\}_{\mathcal{A}}$, then:*

$$S_0 + \mathfrak{P}_l \Sigma \geq \Pi$$

*where $\geq$ applies to each element of the matrices.*

*Proof.* Let $S := S_0 + \mathfrak{P}_l \Sigma$, i.e.:

$$S[b, x] = S_0(x) + \sum_{a \in \mathbf{A}} \mathfrak{P}_l[b, a] \sigma(a, x)$$

Then, $S[b, x]$ represents the state of variable $x \in V$ before one of the occurrences of action $b$ in $l$, because $\mathfrak{P}_l[b, a] = 1$ iff $a$ is before $b$ in $l$. This holds for all occurrences of $b$ in $l$ because there is a row in $S$ for each such occurrence. Since $S_0 + \mathfrak{P}_l \Sigma \geq \Pi$, then $S[b, x] \geq \pi(b, x)$ for all $b$ occurring in $l$ and all variables $x \in V$. $\square$

## Incomparability and Violation in MvPOPs

The result of Lemma 2 is not reversible, i.e., if $S_0 + \mathfrak{P}\Sigma \geq \Pi$, this does not ensure that $\mathfrak{P}$ has only valid linearizations. This happens because some actions could be incomparable, similar to partial order plans, where two incomparable actions w.r.t. the partial order relation can cause a *threat* or a *violation*. This is solved classically by demotion or promotion. A similar phenomenon arises in MvPOPs, but the number of these incomparable occurrences here matters. In the MvTOP case, the set $\lin(\mathfrak{P})$ was a singleton, i.e., no incomparability was allowed and thus no threats. However, if $\lin(\mathfrak{P})$ contains at least two different plans, then at least two occurrences of two actions $a, b \in \mathbf{A}$ are ordered differently in these lists. We call such occurrences *incomparable*.

*In our investment example, in case the initial capital is $c_0 = 7$, $l_1 = [b, b, s, s, g]$, is not valid because, after one occurrence of $b$, the remaining capital is $3 < 4 = \pi(b, c)$, i.e., the two incomparable occurrences of $b$ cause a threat.*

**Definition 7.** *For any MvPOP $\mathfrak{P}$ over $\mathbf{A}$ with max. $g$ and an action $b \in \mathbf{A}$, we say that there are $n$ incomparable occurrences of $a$ to $b$ in $\mathfrak{P}$ (denoted $\mathfrak{I}^{\mathfrak{P}}[b, a] = n$) iff there are at least $n$ occurrences of $a$ in any linearization of $\mathfrak{P}$ ($\mathfrak{P}[g, a] \geq n$), and there are $n + 1$ linearizations s.t. $a$'s occurrences are split into $i$ before $b$ and $n - i$ after $b$ for all $i \in \{0, ..., n\}$.*

**Lemma 3.** *In any MvPOP $\mathfrak{P}$ over $\mathbf{A}$ with max. $g$ and actions $a, b \in \mathbf{A}$, s.t. $a \neq g \neq b$, and $\mathfrak{P}[g, b] > 0$:*
- *$\mathfrak{I}^{\mathfrak{P}}[b, a] = 0$ if $\mathfrak{P}[a, b] > 0$.*
- *$\mathfrak{I}^{\mathfrak{P}}[b, a] = \mathfrak{P}[g, a] - \mathfrak{P}[b, a]$ if $a \neq b$.*
- *$\mathfrak{I}^{\mathfrak{P}}[b, b] = \mathfrak{P}[g, b] - 1$.*

*Proof.* First, if $\mathfrak{P}[g, b] = 0$, $b$ does not occur in any linearization. Second, notice that $\mathfrak{P}[g, a] - \mathfrak{P}[b, a] \geq 0$, because of transitivity. In any linearization, there are $\mathfrak{P}[g, a]$ occurrences of $a$. $\mathfrak{P}[b, a]$ represents the number of $a$'s occurrences before $b$; thus $\mathfrak{P}[g, a] - \mathfrak{P}[b, a]$ are free to occur before or after $b$. Finally, there are $\mathfrak{P}[g, b] - 1$ occurrences of $b$ that are incomparable to each of the other occurrences of that $b$. $\square$

*In $\mathfrak{P}_1$, $\mathfrak{I}^{\mathfrak{P}_1}[b, b] = \mathfrak{P}_1[g, b] - 1 = 2 - 1 = 1$, which is the number of incomparable occurrences of the action $b$ to itself. Remember that the pair $(b, b)$ causes a threat. Intuitively, there is not enough capital to buy twice in a row. We formalize that in the two following definitions.*

**Definition 8.** *For actions $a, b \in A$, we say $a$ violates $b$ in $\mathcal{A}$ (w.r.t. $x \in V$) iff $\pi(b, x) > -\infty$ and $\sigma(a, x) < 0$, and write $a \sim_x b$, as well as $a \sim b$ if such a variable exists.*

**Definition 9.** *For a NAD $\mathcal{A}$, the violation multi-valued relation of an MvPOP $\mathfrak{P}$ over $\mathbf{A}$ with max. $g$ w.r.t. variable $x \in V$ (denoted $\mathfrak{V}_x^{\mathfrak{P}} \in \mathbb{N}^{|\mathbf{A}| \times |\mathbf{A}|}$) is defined for $a, b \in \mathbf{A}$ as:*
- *$\mathfrak{V}_x^{\mathfrak{P}}[b, a] := \mathfrak{I}^{\mathfrak{P}}[b, a]$ if $a \sim_x b$ and $b \neq g$.*
- *$\mathfrak{V}_x^{\mathfrak{P}}[b, a] := 0$ else.*

*Back to our example, there is exactly one incomparable repetition of $b$ to itself, i.e., $\mathfrak{I}^{\mathfrak{P}_1}[b, b] = 1$. Since $b \sim_c b$, we get $\mathfrak{V}_c^{\mathfrak{P}_1}[b, b] = 1$. Therefore, if the initial capital is $c_0 = 7$, in $\mathfrak{P}_1$ (see Fig. 2), accounting for an additional $b$ parallel repetition w.r.t. the variable $c$ results in the state $S$ before applying the second $b$ in $l_1 = [b, b, s, s, g] \in \lin(\mathfrak{P}_1)$, where:*

$$
\begin{aligned}
S(c) &= S_0(x) + \sum_{a \in \{g, s, b\}} (\mathfrak{P}_1[b, a] + \mathfrak{V}_c^{\mathfrak{P}_1}[b, a]) \sigma(a, c) \\
&= 7 + (0 + 1)\sigma(b, c) = 7 - 4 = 3 < \pi(b, c)
\end{aligned}
$$

*Which indicates that $\mathfrak{P}_1$ contains a non-valid linearization $l_1 = [b, b, s, s, g]$. This corresponds to the fact that we cannot buy twice in a row with such initial capital, i.e., that the second occurrence of $b$ is invalid. We can generalize this result.*

In the following theorem, we define sufficient conditions for MvPOP validity by ensuring it contains only valid linearizations. We can use the violation multi-valued relation defined above to account for all violating occurrences of actions to each other and consider the worst-case linearization w.r.t. each variable. Check the supplementary material for the complete proof of this and other theorems.

**Theorem 2.** *For an* NAP $\{S_0 \to g\}_{\mathcal{A}}$*, a multi-set* $(A, \mu)$*, and an MvPOP* $\mathfrak{P}$ *over* $(A, \mu)$ *with max. g:*

$$\text{If for all } x \in V : S_0 + (\mathfrak{P} + \mathfrak{V}_x^{\mathfrak{P}})\Sigma \geq \Pi$$

$$\text{then, } \lin(\mathfrak{P}) \subseteq \{S_0 \to g\}_{\mathcal{A}}$$

## NAP **as Search**

In this section, we use Th. 2 to reformulate the planning problem as a combination of ILP and search. We have already shown that for a fixed number of sequential repetitions, the problem can be solved by ILP. Let us formalize this in a definition.

**Definition 10.** *For a* NAP $\{S_0 \to g\}_{\mathcal{A}}$*, an numeric additive planning fixed problem* $[S_0 \to g]_{\mathcal{A}}^{\mathbf{A}}$ *is defined as the set of linearizations of the MvPOPs* $\mathfrak{P}$ *over a fixed multi-set* $\mathbf{A}$ *with max. g s.t.* $S_0 + (\mathfrak{P} + \mathfrak{V}_x^{\mathfrak{P}})\Sigma \geq \Pi$ *for all* $x \in V$.

Notice that a fixed NAP over a multi-set $\mathbf{A}$ is empty if no MvPOPs over $\mathbf{A}$ with max. $g$ possessing only valid linearizations exist. In the next corollary, we show that solving the NAP by solving fixed NAPs over all possible combinations of sequential repetitions covers all solutions to establish the completeness and soundness of the search.

**Corollary 1** (NAP as Search). *For any* NAP $\{S_0 \to g\}_{\mathcal{A}}$*:*

$$\{S_0 \to g\}_{\mathcal{A}} = \bigcup_{\substack{\mathbf{A}=(A,\mu) \\ \mu:A\to\mathbb{N}}} [S_0 \to g]_{\mathcal{A}}^{\mathbf{A}}$$

*Proof.* From Lemmata 1, 2:

$$\{S_0 \to g\}_{\mathcal{A}} \subseteq \bigcup_{\substack{\mathbf{A}=(A,\mu) \\ \mu:A\to\mathbb{N}}} [S_0 \to g]_{\mathcal{A}}^{\mathbf{A}}$$

And from Theorem 2:

$$\bigcup_{\substack{\mathbf{A}=(A,\mu) \\ \mu:A\to\mathbb{N}}} [S_0 \to g]_{\mathcal{A}}^{\mathbf{A}} \subseteq \{S_0 \to g\}_{\mathcal{A}}$$

$\square$

Corollary 1 can be used to formulate any NAP as a search problem that aims to find the (minimal) number of needed sequential repetitions of each action, i.e., a minimal multi-set $\mathbf{A}$, s.t. an MvPOP $\mathfrak{P}$ over $\mathbf{A}$ containing only valid linearizations exists. We can always find the number of parallel repetitions needed for each action $a \in A$ because we can calculate that using ILP, as shown in Th. 2. Therefore, the undecidability of NAP comes from the hardness of finding the needed number of sequential repetitions in the plan (i.e., $\mu(a)$ for each action $a$), specifically, knowing whether such a number exists or not. To demonstrate the importance of this result, notice that, in general, any planning problem is presented as a path search problem in the states graph from the initial state to one of the goal states. Numeric planning is generally undecidable because the number of states is infinite. Finding an upper bound for the number of needed repetitions of each action makes the graph finite, and thus, the problem becomes decidable in that case. However, in our representation, we only need to find an upper bound for the number of sequential repetitions of each action to guarantee decidability. This will be our approach to finding decidable fragments of numeric planning.

---

**Algorithm 1:** NAP as Search

**Input:** A NAD $\mathcal{A}$, a $g \in A$, and $S_0 : V \to \mathbb{Z}$.
**Output:** A plan $l \in \{S_0 \to g\}_{\mathcal{A}}$.
$\mu : A \to \mathbb{N} : a \mapsto 1$;
**while** *True* **do**
$\quad \mathbf{A} \leftarrow (A, \mu)$;
$\qquad$ Minimize $f(\mathfrak{P})$ s.t. for all $x \in V$:
$$S_0 + (\mathfrak{P} + \lambda\mathfrak{V}_x^{\mathfrak{P}})\Sigma \geq \Pi$$
$\quad$ **if** *There exists MvPOP solution* $\mathfrak{P}$ *over* $\mathbf{A}$ *with max. g for* $\lambda = 1$ **then**
$\quad\quad$ **return** $l \in \lin(\mathfrak{P})$;
$\quad$ **else if** *There exists MvPOP solution* $\mathfrak{P}_r$ *over* $\mathbf{A}$ *with max. g for* $\lambda = 0$ **then**
$\quad\quad$ Analyse $\mathfrak{P}_r$ and update $\mu$;
$\quad$ **else**
$\quad\quad$ **return** "No solution found";

---

**Soundness and Termination of Alg. 1:** Th. 2 proves the soundness of Alg. 1, and Corollary 1 and Lemma 1 show that Alg. 1 terminates for any solvable instance of NAP after less than $|A||l|$ while-loops, where $l$ is the minimal length solution. We solve an ILP with $O(|\mathbf{A}|^2)$ integer variables and $O(|\mathbf{A}||V|)$ constraints in each while loop. However, if $\{S_0 \to g\}_{\mathcal{A}} = \emptyset$, the algorithm may never terminate. The minimization objective $f(\mathfrak{P})$ is irrelevant for the next sections, but we can use it to search for minimal length or cost plans. Furthermore, updating $\mu$ relates to adding sequential repetitions. We will discuss that in the next section.

We use a parameter $\lambda$ to differentiate between the standard and relaxed case. If $\lambda = 0$, then the term $\mathfrak{V}_x^{\mathfrak{P}}$ is removed, and violations are ignored, which makes one sequential repetition of all $a \in \mathbf{A}$ enough for a (relaxed) solution to exist. The algorithm keeps delivering relaxed solutions $\mathfrak{P}_r$ (with $\lambda = 0$) until it finds a valid solution $\mathfrak{P}$ (with $\lambda = 1$). Notice that any valid solution is also a relaxed one. Thus, this relaxation forms an admissible heuristic, extending the relaxed planning graph heuristic (Hoffmann 2003).

**Definition 11.** *For a* NAP $\{S_0 \to g\}_{\mathcal{A}}$*, a relaxed MvPOP solution* $\mathfrak{P}$ *over* $\mathcal{A}$ *of* $\{S_0 \to g\}_{\mathcal{A}}$*, is an MvPOP over* $\mathbf{A}$ *with max. g s.t.* $S_0 + \mathfrak{P}\Sigma \geq \Pi$.

In a relaxed solution, the preconditions of the first repetition of any $a \in \mathbf{A}$ are satisfied in at least one linearization. After analyzing the violations in a relaxed solution, we only need to update the number of sequential repetitions (update $\mu$) when required. Therefore, this heuristic can also help by finding an upper bound for the number of sequential repetitions needed for each action.

*In our investment example with initial capital* $c_0 = 7$*, Alg. 1 would first find the relaxed solution* $l_1 = [b, b, s, s, g]$*, and then notice that no two parallel buying actions can be performed and thus add sequential repetitions of b and s to reach the valid solution* $l_2 = [b, s, b, s, g]$ *which requires indeed two sequential repetitions of b and s. We will show later that this works for any goal profit* $p \in \mathbb{N}$.

# Decidable Fragments of $\mathrm{NAP}$

It is easy to see that any NAP without violations (denoted nvNAP) can be solved directly by ILP since any relaxed solution is valid in that case, i.e., one sequential repetition of each action suffices. Let us first enlarge that NP fragment.

## Maintainability

A threat could emerge whenever two actions $a, b \in A$, with $a \sim b$, are considered incomparable in a relaxed solution. The threat can be solved by promoting $a$ over $b$. However, $a$ could be needed before $b$ as well. Therefore, we might need two sequential repetitions of $a$ to distinguish between occurrences of $a$ before and after $b$. For this reason, we will rely on the relaxed solution to discover action pairs $a \sim b$, where an additional sequential repetition of $a$ is needed.

**Definition 12.** *For a relaxed MvPOP solution $\mathfrak{P}$ over $\mathbf{A}$, we define a binary relation $\precsim_{\mathfrak{P}}$ over $\mathbf{A}$, s.t., for all $a, b \in \mathbf{A}$, $a \precsim_{\mathfrak{P}} b$ iff $a \sim b$, $\mathfrak{P}[b, a] > 0$, and $\mathfrak{I}^{\mathfrak{P}}[b, a] > 0$.*

Notice that $\precsim_{\mathfrak{P}}$ is a partial order over $\mathbf{A}$ because an MvPOP $\mathfrak{P}$ respects irreflexivity, asymmetry, and transitivity. Therefore, $\precsim_{\mathfrak{P}}$ has a maximum for any relaxed solution $\mathfrak{P}$. Only one sequential repetition is required of such a maximum because it does not cause any violation to any action occurring after it. However, this only holds for actions not violating themselves; thus, a distinction is needed.

**Definition 13.** *For an action $a \in A$ in a NAD $\mathcal{A}$, if $a$ does not violate itself (i.e., $a \not\sim a$, or more precisely, $a \not\sim_x a$ for all $x \in V$), we say that $a$ is a maintainable action.*

If there exists a state $S : V \to \mathbb{Z}$ s.t. $S \vDash a$ and $a$ is maintainable, then $a$ can be repeated arbitrarily often from $S$ using one sequential repetition, i.e., any plan $[a, ..., a]$ is valid from $S$. Therefore, in this case, we call $a$ *maintainable*.

*In our investment example, buying, $b$, is not-maintainable. For an initial $c_0 = 7$, $b$ can be applied once but not arbitrarily often. Any repetition after that must be preceded by a sell action, $s$, to collect capital and be able to buy again.*

Now that we know how to repeat a maintainable action $a \in \mathbf{A}$, we can prove that for a relaxed solution $\mathfrak{P}$ over $\mathbf{A}$, if we have upper bounds for all the actions that $a$ can cause a threat to, denoted $T_{\mathfrak{P}}(a) := \{b \in \mathbf{A} : a \precsim_{\mathfrak{P}} b\}$, then, we can also find an upper bound for $a$.

**Theorem 3.** *For a NAP $\{S_0 \to g\}_{\mathcal{A}}$, a maintainable action $a \in A$, and a function $\mu^* : T_{\mathfrak{P}}(a) \to \mathbb{N}$ that defines an upper bound for the number of sequential $b$ repetitions needed for all $b \in T_{\mathfrak{P}}(a)$. We can extend the definition set of $\mu^*$ to $a$ by:*

$$\mu^*(a) := 1 + \sum_{b \in T_{\mathfrak{P}}(a)} \mu^*(b)$$

*Proof.* For any two actions $a, b \in A$, if $a \not\sim b$, the number of sequential repetitions needed of $a$ does not depend on $b$. If $a \sim b$ and $\mathfrak{P}[b, a] = 0$, then, either $\mathfrak{I}^{\mathfrak{P}}[b, a] = \mathfrak{P}[g, a]$ or $\mathfrak{I}^{\mathfrak{P}}[b, a] = 0$. In the first case, all occurrences of $a$ are incomparable to $b$ in $\mathfrak{P}$. Therefore, all $a$ occurrences can be promoted over all $b$ occurrences to solve a threat $(a, b)$. Finally, for $\mathfrak{I}^{\mathfrak{P}}[b, a] = 0$, no ocurrences of $a$ are incomparable to $b$ in $\mathfrak{P}$. Therefore, since $a$ is maintainable, i.e., whenever

$a$ is applicable once, it can be repeated arbitrarily often with one sequential repetition, we conclude that sequential repetitions of $a$ are required only to differentiate between $a$ occurrences after and before each sequential repetition of an action $b$ only if $a \precsim_{\mathfrak{P}} b$. $\qquad\square$

**Definition 14.** *If all actions in a NAP $\{S_0 \to g\}_{\mathcal{A}}$ are maintainable, we call it a maintainable NAP problem (denoted* mNAP*).*

We can characterize mNAP as the fragment of NAP, where a polynomial number of sequential repetitions is sufficient to find solutions for any solvable instance.

**Corollary 2.** *Maintainable NAP is NP-complete.*

*Proof.* First, notice that ILP can be reduced polynomially to a NAP without any preconditions, proving the NP-hardness. Second, if no relaxed solution exists, the problem is unsolvable. Otherwise, given that all actions are maintainable, the upper bounds $\mu^*$ can be calculated iteratively for all actions starting from the maxima in $\mathbf{A}$ w.r.t. $\precsim_{\mathfrak{P}}$. The upper bound for the number of sequential repetitions of any action is at most $\sum_{i=1}^{|A|} i = |A|(|A| + 1)$, i.e., the size of the MvPOP solution is polynomial w.r.t. $\|\mathcal{A}\|$. $\qquad\square$

Notice that the plan length could be exponential w.r.t. $\|\mathcal{A}\|$, if, e.g., the goal $g$ has the precondition $[x \geq 2^n]$, then $n \leq \|\mathcal{A}\|$. If additionally, $\sigma(a, x) = 1$, then, the minimal valid plan $[a, ..., a, g]$ has exponential length w.r.t. $n$. However, we need only $O(n) \subseteq O(\|\mathcal{A}\|)$ bits to encode this plan in an MvPOP, since $2^n$ is an entry in the matrix, that is, an MvPOP can be exponentially more compact than its linearizations.

## Repetition Policies

We can repeat any maintainable action without needing additional sequential copies. This allows us to easily calculate upper bounds for the number of sequential repetitions once a relaxed solution is found. However, this does not apply to non-maintainable actions. In this subsection, we will study how such actions are repeated to avoid irrelevant sequential repetitions during the search. For $a \in A$, let $aA^*a$ denote the set of plans that start and end with $a$. We call a plan $l$ an $a$-repetition if $l \in aA^*a$. E.g., $[a, b, c, d, a]$ is an $a$-repetition. We denote with

$$\sigma(l, x) := \sum_{c \in A} \#_c(l)\sigma(c, x)$$

the total effect of a plan $l$. We will define two types of repetitions that ensure termination for Alg. 1.

**Definition 15.** *An $a$-repetition $l \in aA^*a$ is called beneficial iff for all $x \in V$, if $\sigma(a, x) > 0$, then $\sigma(l, x) > \overline{\sigma(a, x)}$, and $l$ is called de-violating iff for all $y \in V$, if $a \sim_y a$, then $\sigma(l, y) > \overline{\sigma(a, y)}$.*

While searching for the number of sequential repetitions needed of an action $a \in A$, beneficial repetitions ensure that the $a$ effects remain after repeating $a$, i.e., repeating $a$ does not introduce new actions that reverse the effects of $a$. Furthermore, de-violating repetitions ensure that the need for

additional sequential repetitions terminates eventually, i.e., there exists a future point in that search where only parallel repetitions will suffice for the existence of a solution. Let $R(a) := \{S : V \to \mathbb{Z}, \exists l \in aA^*a \text{ s.t. } l \text{ is valid from } S\}$ be the set of states from which $a$ can be validly repeated.

**Definition 16.** *For an action $a \in A$, we call the function $\rho_a : R(a) \to aA^*a$ an a-repetition policy iff for all states $S \in R(a)$, $\rho_a(S)$ is valid from $S$.*

*In our example, the s-repetition $[s, b, s]$ is beneficial: $2\sigma(s, c) + \sigma(b, c) = 2(5) + (-4) = 6 > 5$ and valid from any state $S \in R(s)$. Therefore, increasing the capital $c$ is possible by repeating $s$, and any goal profit $p \in \mathbb{N}$ is reachable by a plan $[b, s, b, s, ..., b, s, g]$. Additionally, the b-repetition $[b, s, b]$ is valid from any state $S \in R(b)$ and de-violating: $\sigma(s, c) + 2\sigma(b, c) = (5) + 2(-4) = -3 > -4$. Consider that from the initial capital $c_0 = 7$ we cannot buy twice in a row, but after buying and selling once, the capital increases by $1$ and two parallel repetitions of $b$ are possible, i.e. $[b, s, b, b, s, s]$ is valid from that initial state.*

**Definition 17.** *For an action $a \in A$, an a-repetition policy $\rho$ is called beneficial / de-violating iff for all $S \in R(a)$, $\rho_a(S)$ is beneficial / de-violating, respectively.*

In this characterization, any maintainable action $a \in A$ has the simplest form of a constant, beneficial and de-violating repetition policy $\rho_a : R(a) \to aA^* a : S \mapsto [a, a]$, which is why one sequential repetition of a maintainable action suffices for arbitrarily many parallel repetitions.

**Definition 18.** *A NAP is called beneficially maintainable (denoted bmNAP) iff all actions have beneficial repetition policies, and finitely maintainable (denoted fmNAP) iff all actions have beneficial or de-violating repetition policies.*

We can use these terms to characterize PP within NAP.

**Lemma 4.** *There exists a polynomial time reduction of PP to bmNAP.*

*Proof.* Remember that we can translate any PP domain to a NAD in polynomial time. Additionally, notice that, in propositional planning, any repetition is beneficial. We can intuitively validate that from our understanding of PP because an activation effect $p$ of $a$ cannot be reversed by an effect $\neg p$ even if it occurs many times before that $a$. E.g., if $\{p\} = \mathit{eff}(a) = \mathit{pre}(g)$, and $\{\neg p\} = \mathit{eff}(b)$, then, after applying the propositional plan $[b, b, b, a, g]$, the effect $p$ of $a$ still satisfies the precondition for $g$ even if $b$ deactivates it arbitrarily many times before. In NAP, this can be done by using the correcting actions $c_p, c_{\neg p}$, e.g., $[b, b, b, a, g]$ in PP translates to $[b, b, c_{\neg p}, b, c_{\neg p}, a, g]$ in NAP. $\square$

In other words, all actions in PP have beneficial repetition policies. This is the second characterization for the difference between PP and NAP we deliver after stating that only sequential repetitions are needed in PP.

Next, we can use induction over the number of non-maintainable actions to prove that the fragment of NAP with the same restriction (bmNAP) is in PSPACE. For that, we show that a bmNAP problem can be decomposed into mNAP problems, which can be solved with polynomial space.

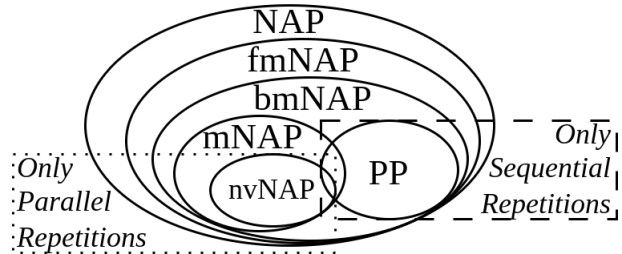

Figure 3: NAP and Repetitions. In ellipses, PP: Propositional Planning, NAP: Numeric Additive Planning, m: maintainable, f: finitely, b: beneficially, nv: non-violating. In rectangles: Types of repetitions needed.

**Theorem 4.** bmNAP *is* PSPACE-*complete.*

Furthermore, by focusing the search on the number of sequential repetitions needed of exactly one action $a \in A$, we get a trace of successive applications of a repetition policy. We can define the sufficient conditions for how such traces can be pruned in fmNAP to prove the following result.

**Theorem 5.** fmNAP *is decidable.*

Check the supplementary material for full proofs of the last two theorems.

## Conclusion and Future Work

We conclude with Fig. 3: Numeric additive planning (NAP) is a superset of propositional planning (PP) where additive effects are allowed; therefore, in NAP, we distinguish between two types of action repetition: (1) Sequential as in PP, and (2) Parallel as in NAP without violations (nvNAP), a superset of ILP. We use search methods for the first and ILP for the latter. Additionally, solving NAP becomes easier when using a compact and least committing plan representation, multi-valued partial order plans, that differentiates structurally between the parallel and sequential repetitions. In general, finding an upper bound for the number of sequential repetitions needed for each action is sufficient for an NAP to be decidable. We can use this observation to study repetition and define mNAP, an NP-complete fragment of NAP, bmNAP, a PSPACE-complete fragment of NAP and a superset of PP, and fmNAP, a decidable superset of both. Finally, we draw a clearer picture of the factors that make NAP undecidable; we characterize these factors as differences to its subsets PP/nvNAP, since only sequential/parallel repetitions are needed for each subset, respectively. Finally, we deliver an algorithm for NAP that always returns a valid solution if it exists and develop heuristics and pruning rules that help decide in many cases if action repetition is irrelevant based on the type of that repetition.

Concerning future work, currently, knowing if an action has a any special kind of repetition policies, e.g. beneficial, is proven to be in PSPACE; we are searching for a tractable characterization of that. Furthermore, we suspect fmNAP to be EXPTime-complete. In general, we believe that developing suitable repetition policies for non-maintainable actions can help discover many more decidable fragments of NAP.

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
