# OpenReview forum: "An Analysis of the Decidability and Complexity of Numeric Additive Planning"
_icaps-conference.org/ICAPS/2024/Conference — ICAPS 2024_

### Official Review · Reviewer_Nxwm · 2024-01-18

**Significance And Importance:** 3
**Soundness:** 4
**Novelty:** 3
**Clarity:** 4
**Overall Evaluation:** 2
**Confidence:** 4

**Weaknesses:**

1: Minor weaknesses that are easily fixable.

**Contributions Of The Paper:**

The paper provides a theoretical analysis of a variant of numeric planning called numeric additive planning (NAP), where action effects can only increase variables and preconditions can test disequalities of variables against constants. The paper studies the problem from an expressiveness and complexity point of view. It first discuss which kinds of preconditions and effects can be encoded into NAP, including classical propositional planning. Then, it analyses the solution plans of NAP problems in terms of the number and the type of repetitions of the actions, and based on this analysis it provides a compact data structure (called multi-valued partial-order plan, MvPOP) to represent plans of NAP problems. Based on this representation of plans, the paper provides a search algorithm for solving NAP problems that uses ILP problems as building blocks. Last, it analyses the complexity of a decidable fragment of NAP showing its NP-completeness.

**Ethical Considerations:**

(1) Not Applicable: The paper does not have any ethical considerations to address

**Nomination For Best Paper:**

No

**Questions For Authors:**

N/A

**Reproducibility:**

0: N/A - nothing to reproduce.

**Strengths Of The Paper:**

The paper is well written, full of complete details and with quite interesting results in the field of the theoretical foundations of planning. The study of tractable numeric planning fragments is important since the formalism is essential in many concrete application domains. It is a solid an interesting contribution to the community and deserves acceptance.

**Weaknesses Of The Paper:**

The introduction provides an extensive discusssion of the different kinds of repetitions of actions in plans, but it lacks a clear remark of what the paper is going to present, until the very end where a single short paragraph is devoted to it. After reading the introduction I had precisely in mind what the paper was *about*, but not what were the original contributions. I suggest the authors to give a bit more space to that paragraph, maybe shortening the (well written) related-work section a bit.

---

> ### Author Rebuttal · Authors · 2024-01-28
>
> Thanks to you and all the other reviewers for taking the time and effort to give us valuable feedback about this research. We will implement the changes suggested in the camera-ready version if the paper is accepted.

---

### Official Review · Reviewer_7XsW · 2024-01-22

**Significance And Importance:** 2
**Soundness:** 3
**Novelty:** 3
**Clarity:** 3
**Overall Evaluation:** 1
**Confidence:** 4

**Weaknesses:**

0: Minor weaknesses requiring some work to be addressed for the paper to be accepted.

**Contributions Of The Paper:**

The paper deals with the analysis of the computational complexity of numeric planning.
First, the paper introduces a novel formalism called NAP that captures the established restricted tasks with integers.
Based on this, a compact encoding of plans is derived, which is based on a matrix-representation of action applications
that allows for partial orders between actions and counts the number of occurrences of every action.
The paper then shows three main complexity results:

1) maintainable NAP is NP-complete, based on a direct ILP encoding:
Here, "maintainable" means that all actions do not disable their own preconditions, so if applicable, can be applied infinitely often in a row.
2) beneficially maintainable NAP is PSPACE-complete:
The "beneficial" part means that repeating an action (possibly with other actions in between) is actually positive, i.e. leads to strictly higher values for all affected variables compared to executing it only once.
3) Finitely maintainable NAP is decidable:
Similar to "beneficial", but non-beneficial actions are allowed if, when repeated with other actions in between, the value of all affected variables with preconditions is strictly greater than after a single execution.

In all cases, the proof constructions are based on computing bounds on the number of times every action must be executed in a plan.

# POST-REBUTTAL:
Thank you very much for the clarifications! I think it would be good to include in particular some more detailed discussion in the related work section.

**Ethical Considerations:**

(1) Not Applicable: The paper does not have any ethical considerations to address

**Nomination For Best Paper:**

No

**Questions For Authors:**

1) How hard is it to check if an beneficial / de-violating repetition policy for an action exists.
2) Is relaxed MvPOP a strict generalization of numeric delete relaxation?
Are there action sets/sequences that are relaxed plans under one, but not the other relaxation?
3) What role does the plan representation play for the complexity results?
4) How does your ILP encoding differ from the one in Helmert (2002)?
5) How does maintainability relate to the causal-graph restrictions of Shleyfman et al (2023)?

**Reproducibility:**

0: N/A - nothing to reproduce.

**Strengths Of The Paper:**

The paper shows some new complexity results for numeric planning that refine the existing theory.
At least some of the results might be promising as a basis for coming up with novel heuristics for numeric planning (depending on Q1/Q2).

**Weaknesses Of The Paper:**

I have two main criticisms:
- Although the paper addresses a known fragment of numeric planning, it does not use established formalisms, but re-defines it in its own terms.
I see that this is beneficial later on to make a better connection to ILP, but it also uses up a lot of space.
The new complexity results only appear on pages 7 and 8, the rest is used to introduce the new formalism (including showing that it's equivalent to established ones) and the matrix-based plan representation.
With that, proofs for two of the main results had to be moved to the appendix.

- The related work section only compares to existing complexity results on a quite high level.
Helmert (2002) already used some ILP to count the number of action occurrences and the corresponding cumulative effects on variables. How does it differ from the proposed approach?
Shleyfman et al (2023) introduces complexity results based on restricting the causal graph. This is very similar to the shown approach (as acknowledged by the authors).
Therefore, some closer comparison would be good to have. It seems like the presented results are more fine-grained, in that they allow cycles in the causal graph, but only if the involved actions are maintainable in some way, i.e. the variables are affected only "positively".


Minor:
- Line 81-84: number references to sections that are not numbered
- line 114: maybe ".. the line of Hoffmann (2003), who proposed.." instead?
- line 115: not sure if ignoring deletes can be considered an "extension of delete lists"
- line 423: there seems to be something wrong here, isn't B(g,b) > 0 a prerequisite?
- Shleyfman,Gnad,Jonsson 2022 vs. 2023: does the workshop version contain separate results; the venue is missing for the 2023 paper
- Venue missing for Scala et al 2016, and "et al" in author list

---

> ### Author Rebuttal · Authors · 2024-01-28
>
> You are correct about redefining many already-known terms. We will try to shorten the introductory section by comparison with ealier work in the camera-ready version.
>
> 1. We found a polynomial space algorithm for finding any type of repetition policy in general (line 719). However, whether there is an easier way to find specific repetition policies is open.
> 2. Yes, in a relaxed MvPOP, we ignore the negative effects of an action that could cause threats if that action is repeated. Ignoring numeric delete lists means ignoring negative effects in general. Therefore, for any delete lists relaxed solution, there exists an equivalent relaxed MvPOP solution.
> 3. The representation allowed to exponentially decrease the bits required to store a solution; therefore, the NP-complete fragment (mNAP) contains planning problems with linear solutions of exponential length (line 615).
> 4. Helmert (2002) proposes a mixed integer program (MIP) to solve numeric planning problems with propositional variables and assigning effects directly. We show that assigning effects are included in NAP by showing that propositional variables are included in NAP; thus, it is equivalent to RT (Th. 1), and using the proof from Schleyfman (2023) that assigning effects are included in RT. Our ILP encoding alleviates the need for distinction between numeric/propositional variables and assigning/additive effects. It partitions the planning problem into repetition and ordering parts. We use MvPOPs for both parts and the violation multi-valued relation (line 439) to account for violations caused by arbitrary repetitions. Helmert's encoding allows us to solve fragments like those without numeric preconditions or those with only positive numeric effects, while our fragments are characterized by the ability to find useful repetition policies.
> 5. In Shleyfman (2022), two variables are causally connected if changing one might require changing the other. The decidable fragment NLRT discussed in that paper considers numeric variables that are leaves in the causal graph. If no propositional variables exist, then all actions in NLRT have a beneficial repetition policy or cannot be repeated. Additionally, we show how all actions with propositional effects can be beneficially repeated (Lemma 4), meaning NLRT $ \subseteq $ bmNAP. However, cycles between variables in the causal graph are not prohibited in bmNAP, i.e., bmNAP $ \not \subseteq $ NLRT. We will mention this in the camera-ready version.

---

### Official Review · Reviewer_yaxp · 2024-01-23

**Significance And Importance:** 2
**Soundness:** 4
**Novelty:** 2
**Clarity:** 3
**Overall Evaluation:** 2
**Confidence:** 5

**Weaknesses:**

1: Minor weaknesses that are easily fixable.

**Contributions Of The Paper:**

The paper explores new decidable and NPC fragments within the generally undecidable simple numeric planning.

**Ethical Considerations:**

(1) Not Applicable: The paper does not have any ethical considerations to address

**Nomination For Best Paper:**

No

**Questions For Authors:**

1. You have a section NAP as Search. Do you intened to implement the proposed methods? It looks like using relaxations on the proposed techniques may lead to powerful heuristics.
2. In Related Work you write "We extend this idea by focusing on the causal structure of actions." And this is the only section where the word  "causal" apears. Can you elaborate on the relation of your methods to causal structures?

**Reproducibility:**

0: N/A - nothing to reproduce.

**Strengths Of The Paper:**

The paper suggests a fresh approach to simple numeric planning, leveraging it to introduce novel decidable and NPC fragments. These fragments have the potential to inspire new heuristics and search techniques, contributing to the advancement of the current state of the art.

**Weaknesses Of The Paper:**

As I denoted in the Weaknesses question there are not major weaknesses, thus I live here a set of comments on the minor things in the paper.

Minor comments:
The sentence below is too cumbersome, and can be divided into two.
Notice that activating p twice instead of once results in the same state, and, ignoring preconditions, any two (propositional) plans of the form [a, b, c], [a, b, b, c] are equivalent, but [b, a, c] is not equivalent to [a, b, c] in general, i.e., repeating an action without
changing the order does not change a propositional plan.
Specifically, I don’t get why do we need this part “but [b, a, c] is not equivalent
to [a, b, c] in general” it does not illustrate what comes after the i.e.

I don’t get why the next paragraph starts with the word “Conversely”.

I think that for parallel execution you need both associativity and commutativity.


The related work effectively introduces the reader to the work that has been done so far. However, the paper fails to explicitly reference these relationships (some of wich are indicated below) within its main content.

NAD seems to be equivalent to the RT Hoffmann (2003) defined for the interval relaxation, with only difference that the propositional variables were removed, and the goal action introduced.

In definition 3 something went wrong with the indices “if i ≥ 1, for all i ∈ {0, .., n}.”

You should probably either bold or italicize the definition of the planning problem. Moreover, it feels like a planning problem should be defined prior to a plan.

In the Expressiveness subsection, the trick on <, \leq precondition is exactly what is done for RT in Hoffmann (2003). You should cite the paper.

Note in this subsection you use interchangeably >, < and \geq, \leq signs. Obviously, >,< can be translated to  \geq, \leq  over integers, but you should probably mention this.

I like the trick in the Effects. Note that it can make the plans longer by a factor of |P|, if I’m not mistaken.

After definition 5, the plan linearizations may be exponential in size, if I’m not mistaken. This probably should be mentioned.

In defintions 8 and 9, shouldn't you mentione Coles et al. (2008)?

It seems that NAP without violations is equivalent to interval relaxation as it was defined by Hoffman (2003).

Corollary 2. can bee seen as an extension of the proof the numeric h^{+} is npc (see Kuroiwa et al. 2021)

In the proof of Lemma 4 you use \textit{pre} and \textit{eff} that were not previously defined.


The definition of the multi-sets of actions \textbf{A} = (A, \mu) is confusing.

Here are commands for += and -=:
\newcommand{\pleq}{\mathrel{{+}{=}}}
\newcommand{\mineq}{\mathrel{{-}{=}}}
You can also use \models instead of \vDash.
This is obviously a matter of taste.

Now, correctly me if I’m wrong, but if I know the bound on the number each action can be applied, denote it #a and I know the effect of this action on the variable x, denote it \sigma(a,x). Then the domain of x is bounded by |x| \leq \sum_{a \in A} #a |\sigma(a,x)|. Thus, the domains of all numeric variables are bounded, and the problem is decidable.


*POST-REBUTTAL*
===============
Thank you for your response. I believe adding the clarifications you pointed out should improve the paper.

---

> ### Author Rebuttal · Authors · 2024-01-27
>
> You are correct. Both associativity and commutativity are needed. We will update that.
>
> Th. 1 proves that NAD and RT Hoffmann (2003) are equivalent.
>
> You are correct about the trick taken from Hoffmann. We mention that in general (line 249).
>
> You are correct: Using the correcting actions trick (line 230), a propositional plan of length $ n $ is translated into a numeric plan with length $ \leq n + |P|n $.
>
> We mentioned that a linearization can have exponential length w.r.t. the MvPOP size (line 615).
>
> We mention (Coles et al. 2008) when speaking of repetition types (line 119). However, we should have compared our repetition types to theirs.
>
> You are right about non-violating NAP being equivalent to interval relaxation (Hoffmann 2003). We mentioned that vaguely (line 117).
>
> We mention that bounding action repetition generally makes the problem decidable (line 491). However, we extend that result by proving that bounding sequential repetitions is enough for the problem to be decidable (line 498).
>
> 1. We are working on implementing the algorithm and testing it against other known planners for benchmark problems.
> 2. With "causal structure of actions", we mean the violation relation between actions, i.e., that repeating one arbitrarily often affects the preconditions of the other. This should have been clarified in the paper.

---

### Meta-Review · Area_Chair_Z1eC · 2024-02-05

**Recommendation:** Accept (Oral)
**Confidence:** 4

**Metareview:**

After the discussion, all reviewers agreed to accept the paper. The one reviewer that was initially more critical of the contribution raised their evaluation after seeing the rebuttal. Because the discussion quickly converged on acceptance, there wasn't much further discussion.

In the discussion we had, one of the reviewers emphasized again that they would like to see the connections to related work made clearer, with more detail provided. With an increasing number of nontrivial complexity results in the literature, for future readers it is critical that the (at times quite nuanced) similarities and differences are very explicitly delineated. There was agreement on this point.

There was also some discussion regarding the redefinition of the numeric planning formalism compared to earlier work. One reviewer disliked that a new avenue was taken here; another reviewer particularly liked the redefinition and considered it an important contribution, as new formalizations provide new perspectives. (I agree with this point.) In any case even the reviewer that didn't like this aspect didn't see it as a blocker.

**Ethical Considerations:**

(1) Not Applicable: The paper does not have any ethical considerations to address